# A Competitive Panning Method Reveals an Anti-SARS-CoV-2 Nanobody Specific for an RBD-ACE2 Binding Site

**DOI:** 10.3390/vaccines11020371

**Published:** 2023-02-06

**Authors:** Siqi He, Jiali Wang, Hanyi Chen, Zhaohui Qian, Keping Hu, Bingjie Shi, Jianxun Wang

**Affiliations:** 1School of Life Sciences, Beijing University of Chinese Medicine, Beijing 100029, China; 2NHC Key Laboratory of Systems Biology of Pathogens, Institute of Pathogen Biology, Chinese Academy of Medical Sciences and Peking Union Medical College, Beijing 100730, China; 3The Institute of Medicinal Plant Development, Chinese Academy of Medical Sciences and Peking Union Medical College, Beijing 100730, China; 4Andes Antibody Technology Hengshui LL Company, Hengshui 053000, China; 5Shenzhen Research Institute, Beijing University of Chinese Medicine, Shenzhen 518118, China

**Keywords:** SARS-CoV-2, ACE2, phage display, nanobody, biopanning

## Abstract

Most neutralizing antibodies neutralize the severe acute respiratory syndrome coronavirus 2 (SARS-CoV-2) by directly blocking the interactions between the spike glycoprotein receptor-binding domain (RBD) and its receptor, human angiotensin-converting enzyme 2 (ACE2). Here, we report a novel nanobody (Nb) identified by an RBD-ACE2 competitive panning method that could specifically bind to the RBD of SARS-CoV-2 with a high affinity (EC_50_ = 0.03 nM) and successfully block the binding between the RBD and ACE2 recombinant protein. A structural simulation of the RBD-VHH complex also supports a mechanism of the Nb to block the interaction between the RBD and ACE2. A pseudovirus assay of the Nb showed it could neutralize the WT pseudovirus with high potency (IC_50_ = 0.026 μg/mL). Furthermore, we measured its binding to phages displaying RBDs of different SARS-CoV-2 variants and found that it could bind to recombinant phages displaying the RBD of beta and delta variants. This study also provides a method of phage library competitive panning, which could be useful for directly screening high-affinity antibodies targeting important functional regions.

## 1. Introduction

After the epidemics caused by the severe acute respiratory syndrome coronavirus (SARS-CoV) and the Middle East respiratory syndrome coronavirus (MERS-CoV), the third novel pneumonia outbreak caused by the severe acute respiratory syndrome coronavirus 2 (SARS-CoV-2) [1] has persisted globally for nearly three years. Due to the fact that SARS-CoV-2 is a single-stranded RNA virus with a high mutation rate [2], a large number of SARS-CoV-2 variants have appeared continuously in a relatively short period of time during its transmission (https://www.who.int/en/activities/tracking-SARS-CoV-2-variants/, accessed on 23 February 2022), posing a great threat to public health. At present, the cumulative number of confirmed coronavirus cases worldwide exceeds 614 million, and the number of deaths exceeds 6.5 million (https://covid19.who.int/, accessed on 5 June 2022). Although various kinds of vaccines have been widely promoted in some countries, preventing the spread of COVID-19 remains a challenge that will persist for a long time.

SARS-CoV-2 is a single-stranded positive-sense RNA-enveloped virus. Its membrane surface contains many spike proteins (S proteins for short), which form the crown-like appearance of SARS-CoV-2. The S protein is in fact a homotrimeric protein [3], and each protein monomer contains one S1 subunit (14–685 aa) and one S2 subunit (686–1273 aa). The S protein can recognize and bind the angiotensin-converting enzyme 2 (ACE2) receptor on the surface of the cell membrane [4], and after being cleaved by the serine protease TMPRSS2 on the cell surface, S mediates the fusion of the viral envelope and cell membrane, promoting the viral RNA genome entry into the host cell [5]. Among the S protein regions, the receptor-binding domain (RBD) of the S1 subunit is a key region for the binding between SARS-CoV-2 and ACE2 receptors and antibody recognition, and is considered the most effective target of anti-SARS-CoV-2-neutralizing antibodies (NAbs) to date [6,7,8].

The global outbreak of the coronavirus pandemic highlights the need to rapidly develop effective methods for the treatment and prevention of SARS-CoV-2 infection. Nab therapy is a beneficial weapon for managing the major public health crisis. This promising treatment has attracted much attention and expectation because it can achieve the dual effect of prevention and treatment. NAbs usually work by binding with the viral RBD, thus preventing the virus from being adsorbed on the ACE2 receptor and abrogating viral penetration into cells for replication and proliferation. In addition, NAbs form immune complexes with viruses, and these complexes are easily engulfed and cleared by macrophages [9]. During the previous outbreaks of SARS and MERS, many developed monoclonal antibodies (mAbs) with virus-neutralizing activity showed great potential in the treatment of coronavirus infection [10,11,12]. However, the clinical application of mAbs has been hindered by their expensive and time-consuming manufacturing process in eukaryotic systems.

An attractive alternative to mAbs is the single-domain antibody (sdAb) of camel immunoglobulin, which was first discovered in 1993 by the Belgian scientist Hammers Casterman and coworkers [13]. It is a natural antibody that lacks light chains and contains only a heavy chain variable region and two conventional CH2 and CH3 regions. The sdAb binds to the antigen through the variable region on its heavy chain. This variable region can exist independently and stably in vitro. It is called the variable domain of the heavy chain of heavy-chain antibody (VHH) or nanobody (Nb), with a molecular weight of 12–15 kDa, and is the smallest segment known to bind to the antigen [14]. Compared with the traditional mAb (150 kDa), this antibody further has multiple beneficial features, such as more available epitopes, relatively low production cost, stronger tissue penetration ability, and easier production in prokaryotic expression systems [15].

Phage display technology was first created by Professor George P. Smith in 1985 [16] and is the preferred strategy for Nb screening. This technology uses genetic engineering methods to insert foreign gene fragments into the genome of a bacteriophage so that the protein or polypeptide encoded can form a fusion protein with the bacteriophage capsid protein, which can be displayed on the surface of the bacteriophage. The displayed protein can maintain its relative spatial structure and biological activity, so the target protein can be used as a “bait” to fish out the target protein that binds to it, which can be used for screening a bacteriophage library [17].

Based on the phage display method, the excellent biological characteristics of VHHs, and the basic principle of RBD binding to ACE2 to mediate virus entry into host cells, the main purpose of this study was to use the phage VHH library successfully constructed in the early stage of our laboratory to screen VHHs with wild-type SARS-CoV-2 RBD as the target antigen by using a method of RBD-ACE2 competitive panning and to express and identify the screened VHH to find a high-affinity VHH specific to the SARS-CoV-2 RBD recombinant protein, providing an idea for the development of neutralizing nanobodies. In addition, the combination of the VHH with the recombinant phage displaying the RBD of SARS-CoV-2 variants will be applied to preliminarily determine the binding ability of the VHH to several key mutants of the SARS-CoV-2, thus laying a solid foundation for further research on the screening of VHHs against new coronavirus mutants.

## 2. Materials and Methods

### 2.1. SARS-CoV-2 RBD-ACE2 Competitive Planning of Phage Libraries

The VHH library (not yet published) was constructed after immunizing llamas with SARS-CoV-2 WT RBD recombinant protein and was used in the biopanning experiment. The capacity of this library is up to 4 × 10^12^ pfu/mL and has high diversity, which is thought to be beneficial for screening for bioactive VHHs against RBD target proteins.

The phage library was panned by solid-phase panning (Figure 1), which is the most widely-used antigen presentation strategy [18]. Based on previously described methods [19], we applied an optimized panning strategy that can be used to identify antibodies with certain binding characteristics. A high-binding polystyrene flat-bottom microplate (#42592, Corning, NY, USA) was coated overnight at 4 °C with 5 µg/mL SARS-CoV-2 RBD-Fc (#40592-V02H, Sino Biological, Beijing, China) or coating buffer (pH 9.6) as a control. Two rounds after the first round of preliminary RBD screening, recombinant human ACE2-His protein (#10108-H08H, Sino Biological, Beijing, China) was added for competitive panning. After blocking wells with blocking buffer (2% BSA in PBST, pH 7.4) for 2 h, we added phages that were preincubated in blocking buffer for 1 h at 37 °C to the wells and incubated the plates with coating antigen at 37 °C for 2 h. Unbound or loosely bound phage particles were washed out by extensive washing with PBST (PBS buffer containing 0.05% Tween 20, pH 7.4), and bound phages were then eluted by incubation with 100 µg/mL trypsin (Solarbio, Beijing, China). Eluted phage particles were immediately supplemented with log-phase *Escherichia coli* (TG1) cells. The infected TG1 cells were used either to amplify the phages used as input for the next round of panning or to titre the phages for determining the enrichment of each round.

In the second and third rounds of panning, 2 µg/mL ACE2 was pre-incubated with the coated RBD before adding phages, and the subsequent steps were the same as those described above. To enrich the tight binders, the stringency of the washing conditions used in the rounds of panning was gradually increased. In the second and third rounds, the concentration of coated RBD was reduced to 1 µg/mL and 0.5 µg/mL, and the concentration of competition antigen ACE2 in the third round was 1 µg/mL. Moreover, the concentration of Tween 20 and the number of washes were consecutively increased in each round of panning [20]. A 0.05% PBST solution (washing 10 times, for 30 s each) was used to wash the weak binders of SARS-CoV-2 RBD away during the first round, whereas 0.1% (washing 20 times) and 0.2% (washing 30 times) PBST wash solutions were used for the second and third rounds of panning.

### 2.2. Phage ELISA for Identification of Positive Clones

The specific binding of phages to the RBD was tested by phage enzyme-linked immunosorbent assay (ELISA) [21]. After each panning, the resultant polyclonal phage pools were tested; finally, the third-round output phage pools infecting *E. coli* TG1 were chosen for further analysis. The moderately-diluted bacterial culture was plated, and 24 individual colonies were randomly picked and expanded for monoclonal phage production. Phage pools or monoclonal phage were added to a pre-blocked RBD-coated 96-well plate (0.5 μg/well) and, after being washed 15 times with PBST, were developed by HRP-conjugated anti-M13 MAb (diluted 1/8000 in 2% BSA/PBS). After 1 h of incubation, the plates were rinsed ten times, and detection was carried out by adding tetramethylbenzidine (TMB) as a chromogenic substrate. The reaction was stopped with 1 M HCl, and the absorbance was measured at 450 nm. Wells with only coating buffer were included as controls to test for nonspecific binding. After the monoclonal phage ELISA, 10 positive clones with higher A450 values were selected, and part of the bacterial culture was used for sequencing.

### 2.3. Competitive Phage ELISA

RBD-Fc (0.2 μg/well) was coated in a microplate in 100 μL of coating buffer (pH 9.6) at 4 °C overnight. Then, 100 μL/well 2% BSA-PBST was incubated for 1.5 h for blocking. After washing 3 times, a serial phage solution was added and incubated with RBD-Fc for 1 h. Anti-M13/HRP conjugate (diluted 1/10,000 in 2% BSA/PBS dilution) and TMB were used to amplify the signals and to develop color, respectively. After stopping the reaction via 1.0 M HCl, the absorbance was measured at 450 nm. After that, the sub-saturation (80% of maximal effect) concentration of the phage solution was used for competitive ELISA.

RBD-Fc (0.2 μg/well) was coated for competitive phage ELISA. After blocking for 1.5 h and washing 3 times, the sub-saturation concentration of VHH-phage solutions was mixed with 4 μg/mL ACE2-His (SinoBiological). The mixture of phage and ACE2-His was added to wells coated with RBD-Fc and incubated for 1 h, and the following steps were as described previously.

### 2.4. Prokaryotic Expression and Purification of the VHH Obtained by Screening

We used Gibson assembly cloning to integrate the VHH gene into the pET22b+ expression vector (HedgehogBio, Shanghai, China) containing a C-terminal histidine affinity tag with flanking NcoI/XhoI restriction sites. Following the transformation of the recombinant vector into *E. coli* DH5α (TIANGEN, Beijing, China), positive clones were identified, the culture was expanded, and the plasmid was transformed into the *E. coli* BL21 (DE3) expression host (Solarbio, Beijing, China). A fresh single colony harbouring the plasmid was picked and cultivated in Luria-Bertani (LB) medium supplemented with 100 µg/mL ampicillin. Expression was induced by the addition of 1 mM isopropyl-beta-d-thiogalactopyranoside (IPTG) (Sigma, St. Louis, MO, USA) and allowed to continue for 6 h at 30 °C. Cells were harvested by centrifugation and lysed by sonication of the cell pellet. The protein was purified from the supernatant by immobilized metal affinity chromatography (IMAC) using nickel–nitrilotriacetic acid (Ni-NTA) agarose resin (#17531806, Cytiva, Marlborough, MA, USA) and then concentrated and diluted in PBS with a 3KD Microsep Advance Centrifugal Device (#MCP003C46, PALL, PWA, NY, USA). The size and purity of the VHH were analysed by reducing dodecyl sulfate polyacrylamide gel electrophoresis (SDS–PAGE). The concentration of the VHH was determined according to the BCA method using BSA as a standard.

### 2.5. Validation of VHH Binding Specificity to the SARS-CoV-2 RBD by ELISA

The SARS-CoV-2-RBD and BCMA recombinant proteins at 0.5 µg/well were coated overnight at 4 °C in triplicate on a polystyrene flat-bottom microplate, and wells with only coating solution served as blank controls. After washing three times with PBST to prevent nonspecific binding, we blocked the plate using blocking buffer for 2 h and then washed the plate at room temperature (RT). VHHs at a concentration of 0.5 µg/mL were added to each well and incubated at 37 °C for 1 h. The plate was then washed, and an HRP-conjugated anti-His antibody (diluted 1/5000 in 2% BSA/PBS) was added and incubated at 37 °C for 1 h in the dark. The process of substrate addition, termination, and plate reading described for the ELISA was repeated.

### 2.6. Determination of the Binding Capacity of the VHH to the SARS-CoV-2 RBD (EC_50_)

The median effect concentration (EC_50_) of VHH binding to the SARS-CoV-2 RBD was measured with ELISA. After blocking wells, we added 5-fold serially diluted VHH (from 0.00256 to 200 ng/mL) to RBD-coated (0.5 µg/well) microtiter plates and incubated the plates at 37 °C for 1 h in the dark. The samples were analysed following the assay procedure described above.

### 2.7. Validation of VHH Competition with ACE2 for Binding to the SARS-CoV-2 RBD

Next, 0.5 µg/mL RBD solution was added to each well, and the microplate was coated at 4 °C overnight. The plate was then washed and blocked with 2% BSA for 2 h in a 37 °C incubator. Fifty microlitres of VHH solution (1, 0.5, 0.2, 0.1 µg/mL) was mixed well with 50 μL of biotinylated ACE2 (#10108-H08H-B, Sino Biological, Beijing, China) solution (1.6 µg/mL), and an equal volume of VHH solution mixed with PBS was used as a control. After washing the wells, we added 100 μL of the above mixture to each well, and the three components were fully combined at 37 °C for 2 h. The washing process was repeated, and 100 µL of HRP-labelled streptavidin (diluted 1/5000 in 2% BSA/PBS) was added to each well, followed by incubation of the plate at 37 °C for 1 h. The remaining steps were performed as described above.

### 2.8. Docking Simulation Studies and Analysis of the Complexes

To explore the structural basis for the VHH blocking the interaction between the SARS-CoV-2 RBD and ACE2, the structures of the VHH and RBD complexes were simulated. A homology model of VHH_5-05_ was built on the SWISS-MODEL online server (https://swissmodel.expasy.org, accessed on 12 October 2022) using the searched templates. The simulation was based on the structure of hACE2 with the SARS-CoV-2 S RBD (PDB: 6LZG) [22]. We performed molecular docking to simulate protein interactions by MOE software, and the Protein–Protein Dock module in the DOCK package was used to realize the docking calculation between RBD and VHH_5-05_. In this module, the parameter was set to select the antibody CDR region, 10,000 Pre-Placement, and 1000 Placement, and finally, the models were output with the top 100 Refinement. After obtaining the combination of simulations, Ligplot+ was further used for 2D visualization [23].

### 2.9. Pseudovirus-Based Neutralization Assay

To determine the neutralization activity of VHH_5-05_, the pseudotyped virus neutralization assay was performed. Briefly, 293T cells were co-transfected with psPAX2, pLenti-GFP, and plasmids encoding either SARS-CoV-2 S by using polyetherimide (PEI), as previously described [24]. The supernatants containing SARS-CoV-2-pseudotyped virus were harvested.

Pseudovirions were pre-incubated with serially-diluted VHH_5-05_ for 1 h in 37 °C, then virus-VHH mixture was added onto 293/hACE2 cells (10^4^ per well) in a 96-well plate. About 40 h post inoculation, cells were collected and lysed with luciferase substrate (Bright-Lumi™ Firefly Luciferase Reporter Gene Assay Kit, #RG051M, Beyotime, Shanghai, China) at room temperature for 5 min, then the luciferase luminescence (RLU) of each well was measured with a luminescence microplate reader (SpectraMax i3x Microplate Reader, Molecular Devices, CA, USA). Half maximal inhibitory concentration (IC_50_) value was determined by a four-parameter non-linear regression model using PRISM. All experiments were carried out in triplicates and repeated at least twice.

### 2.10. Binding Activity of VHH to Recombinant Phage Displaying the RBD of SARS-CoV-2 Mutants

Using the antibody detection method based on phage display technology that was successfully constructed in our laboratory previously [25], the binding activity of the screened VHH_5-05_ to the recombinant phage displaying the SARS-CoV-2 mutant RBD was preliminarily verified. The recombinant M13KO7 phage containing the RBD fragments of each mutant strain of SARS-CoV-2 was prepared, and purified VHH_5-05_ was coated on a high-adsorption microplate overnight at 4 °C. The recombinant phage of each mutant strain RBD was added to bind VHH_5-05_, unbound phages were washed away, bound phages were thermally lysed, and then the gene fragment of those phages were amplified by qPCR to indirectly reflect the specific binding of VHH_5-05_ to the recombinant phage. Measurements were made in triplicate.

### 2.11. Quantification and Statistical Analysis

The results of the polyclonal phage ELISA and binding experiments are the means of three independent experiments. All data were graphed using Prism software (version 8, GraphPad Software).

## 3. Results

### 3.1. Enrichment of ACE2-Competitive Tight Binders against the SARS-CoV-2 RBD by Panning

The mechanism of action of SARS-CoV-2-neutralizing antibodies is usually based on blocking the interaction of S/RBD with the specific receptor ACE2, either directly or through steric hindrance. Due to its small size, the VHH may not effectively sterically block the binding of antigens to their receptors, as traditional mAbs do [26]. Therefore, we used competitive panning methods to drive the selection of VHHs that may act against the important functional region—epitopes shared between the RBD and ACE2.

To remove target-unrelated phages that bound directly to the flat-bottom microplates, the phage library was incubated directly with an empty well of the microplate before biopanning. The first round of panning eliminated those phages that did not bind or bound weakly to the RBD. With the addition of ACE2 and the more stringent panning conditions in the subsequent steps, a total of three rounds of panning were carried out. These iterative rigorous panning procedures resulted in the successful enrichment of high-affinity VHHs. The recovery efficiency was estimated by dividing the output phage titres by the input phage titres, as presented in Table 1. The recovery efficiency of each round was calculated as the output titre divided by the input titre [27].

### 3.2. Selection of SARS-CoV-2 RBD-Specific Binders

The output phages derived from the final round of biopanning were used to evaluate the specific binding of individual clones to the SARS-CoV-2 RBD. A total of 24 single colonies on the titre plate were randomly picked, inoculated, and cultured to prepare phage monoclonal supernatant and identified by phage ELISA. The ELISA results of polyclonal phages and the 21 successfully prepared monoclonal phages are shown in Figure 2. We selected 10 monoclonal samples with higher absorbance values for sequencing. A multiple sequence alignment analysis of the amino acid sequences showed that the obtained antibodies all conformed to the typical camel-derived heavy chain sequences. MEGA phylogenetic tree analysis of sequencing results showed that the sequences are diverse (Figure 3). 

In direct immobilization experiments, the coated antigen has an altered native conformation, which may lead to the discovery of antibodies that recognize only the altered conformations of the antigens [28]. Therefore, the non-immobilized SARS-CoV-2 RBD was used for competitive ELISA. As expected, the binding of VHH-displaying phages to immobilized antigen was obviously diminished in the presence of a correctly folded SARS-CoV-2 RBD, confirming that the screened VHH could recognize the antigen in its native conformation.

### 3.3. VHH-Phages with RBD-ACE2 Blocking Capability

Among the 10 monoclones corresponding to those randomly selected by phage ELISA, 4 showed the ability to block the binding of RBD to ACE2 (Figure 4). We selected the VHH A5 clone with the strongest blocking ability for further prokaryotic expression and identification, referred to as VHH_5-05_.

### 3.4. Expression and Purification of VHH5-05

A prokaryotic expression system and IMAC were used to produce the target His-tagged Nb. Briefly, the protein was efficiently produced by using a pET 22b+ expression vector containing the VHH_5-05_ gene in an E. coli BL21 (DE3) host and further purified using a Ni-NTA agarose column. Protein expression and purification were assessed by 4–20% (*w*/*v*) SDS–PAGE using the method described by Laemmli [29], and protein bands were stained with Coomassie Brilliant Blue R250.

SDS–PAGE analysis of expression samples indicated that the supernatant contained sufficient amounts of a soluble target protein (Figure 5a). The supernatant was submitted to further purification and elution to obtain a high-purity target protein with a molecular weight of approximately 14 kDa (Figure 5b), which was consistent with the theoretical molecular weight of the VHH.

### 3.5. Binding Specificity and Affinity Determination of VHH5-05 to the SARS-CoV-2 RBD

The binding specificity of the purified VHH to the SARS-CoV-2 RBD antigen was verified by ELISA. The results showed that the VHH obtained by panning could specifically bind to the SARS-CoV-2 RBD (Figure 6a). The binding affinity of the obtained VHH to the SARS-CoV-2 RBD was determined by the ELISA saturation concentration method, and the EC_50_ of the obtained VHH was 0.03 nM (Figure 6b), indicating that the RBD-ACE2 competitive panning method used in this study successfully identified a camel-derived VHH with higher affinity.

### 3.6. VHH_5-05_ Competes with ACE2 for Binding to the SARS-CoV-2 RBD

After mixing purified VHH_5-05_ with the SARS-CoV-2 RBD, the competition with ACE2 was tested. The results showed that the VHH could block the interaction between human ACE2 (hACE2) and the RBD (Figure 6c); that is, VHH_5-05_ occupied the RBD and ACE2 binding site, allowing ACE2 that was no longer bound to be removed during plate washing. At 0.5 µg/mL, VHH_5-05_ showed significant blockade of binding, and the effect remained significant at a concentration of 0.1 µg/mL, further confirming the feasibility of the competitive panning method in this study. Based on the results of this verification step, it can be preliminarily inferred that the screened VHH_5-05_ has substantial overlap with the binding sites of ACE2-RBD.

### 3.7. Prediction of VHH_5-05_ Binding Sites to RBD Using a Docking Simulation

The accession number of the S protein sequence of SARS-CoV-2 is YP_009724390.1, and the receptor-binding motif (RBM) of SARS-CoV-2 is from 437 to 508 aa. The key amino acids in hACE2 that interact with the RBM are K31, E35, D38, M82, and K353 [30]. Those amino acids corresponding to SARS-CoV-2 are L455, F486, Q493, S494, N501, and Y505 [31], and all molecules are shown in cartoon representation as in Figure 7a. According to the analysis of the structure prediction composite obtained from the docking of the RBD-VHH_5-05_ (Figure 7b), the molecular docking fraction was −78.1905, indicating that VHH_5-05_ could bind well to RBD. A 2D graph (Figure 7c) of the protein molecule interaction displayed by Ligplot+ intuitively showed the interaction forces, including hydrogen bonds and hydrophobic interactions, indicating that the simulated conformational binding was strong. The modeling analysis result revealed that most residues on the RBD-VHH epitope overlapped with the RBD-ACE2 binding interface, specifically, the key sites—F486, Q493, and S494 of the SARS-CoV-2 RBD participate in its binding to VHH_5-05_, which confirms why the Nb can effectively block the binding of RBD and ACE2.

### 3.8. Pseudovirus Neutralization Ability of VHH_5-05_

Of VHH_5-05_, 1:3 8-point serial dilutions were prepared in medium and pseudovirus was added 1:1 to 110 μL of each antibody dilution. In regards to the pseudovirus, VHH mixtures were incubated for 1 h at 37 °C and then mixed with 10^4^ 293/hACE2 cells for 48 h. After lysing cells with luciferase substrate and collecting RLU values, IC_50_ value was calculated by fitting the RLU to a sigmoidal dose–response curve. The results showed that VHH_5-05_ could neutralize the WT pseudovirus with high potency, as the IC_50_ was 0.026 μg/mL (Figure 8).

### 3.9. Binding of VHH_5-05_ to Recombinant Phages Displaying the RBD of SARS-CoV-2 Mutants

According to the qPCR results, the specific binding of the coated VHH_5-05_ to the recombinant phage displaying the RBD of the SARS-CoV-2 mutant strain was analyzed to preliminarily determine the binding ability of the VHH_5-05_ to the RBD of each mutant strain. The VHH obtained in this study not only showed strong binding ability to phages that displaying wild-type RBD but also showed binding to phages displaying the beta mutant RBD and delta mutant RBD (Figure 9), which provides a theoretical basis for further verification of the neutralizing ability of the VHH_5-05_ against these two mutant strains. However, VHH_5-05_ showed poor binding to phages displaying the omicron mutant RBD, as we anticipated. Similar results were obtained in our binding assay with recombinant omicron B.1.1.529 protein, which is consistent with the conclusion in many studies [32,33,34] that the SARS-CoV-2 omicron mutants escape most of the existing NAbs.

## 4. Discussion

Since the pneumonia epidemic caused by the SARS-CoV-2 was officially reported at the end of 2019, it has transmitted globally in a short time. Since its human-to-human infectivity is much higher than that of other coronavirus outbreaks [35], the World Health Organization (WHO) declared the SARS-CoV-2 pandemic a public health emergency of international concern at the beginning of 2020. The rapid mutation and clever concealment of this coronavirus have affected the continually developed lines of defence; as soon as one problem is addressed, a new challenge arises. It takes a long time to develop traditional vaccines and small-molecule drugs; regarding plasma therapy, the source of plasma is limited, the effect is uncertain, and it is difficult to apply in practice. Among the numerous therapies to prevent and treat COVID-19, the advantages of NAbs have been gradually highlighted. A Nab is an antibody that has the dual effects of prevention and treatment against SARS-CoV-2, and such antibodies have been employed after screening, preparation, and validation. NAbs have a single composition and good safety and can accurately target the antigen site. In 2018, Zhao et al. [36] developed a new neutralizing Nb (NbMS10) and its human Fc fusion version (NbMS10 Fc) targeting the MERS-CoV spike RBD, both of which bind to the MERS-CoV RBD with high affinity; in particular, the half-life of NbMS10 Fc in vivo is significantly prolonged and can completely protect humanized mice from lethal MERS-CoV attack, and a single-dose treatment shows high prevention and treatment efficacy. The excellent NAb DXP-604 [6,37], jointly developed by Xie et al. of Peking University, was approved for use as a “compassionate emergency treatment drug” for SARS-CoV-2 patients in Beijing Ditan Hospital in 2021, and it achieved good therapeutic effects.

In the treatment strategy for viral pneumonia with the respiratory tract and lung as the main infection sites, the use of Nbs has unique advantages. Owing to the benefits of the excellent physical and chemical properties of Nbs, such as their small molecular weight and stability at room temperature, it is possible to effectively atomize and locally administer drugs to improve the drug concentration in the respiratory tract, alveoli, and other viral infection niduses [38]. Researchers from Fudan University reported the broad-spectrum bispecific all-human VHH bn03 for use against SARS-CoV-2 [39]. As an aerosol inhalation preparation, the antibody showed good efficacy in mouse models of mild and severe SARS-CoV-2 infection. The bispecific Nb (bi-nanobody) has entered the pilot production stage, and its progress to clinical trials is accelerating.

Antibody drug development is the most remarkable achievement of phage display technology. Phage display combines genotype and phenotype, while panning of phage display antibody libraries simulates the natural selection process [40]. By flexibly adjusting the panning pressure, we can obtain more antibodies with stronger affinity or certain characteristics from the phage library [41]. To maximize the chance of discovering Nbs with the desired characteristics of targeting the SARS-CoV-2 RBD-ACE2 binding epitope, we devised a new panning strategy—ACE2 protein was added to compete with VHH-expressing phages to bind the coated solid SARS-CoV-2 antigen. Successfully, the phage VHH library was enriched after three rounds of panning, and the purified Nb binds the SARS-CoV-2 RBD with high affinity and can compete with ACE2 for RBD binding. Furthermore, we found that the VHH also showed significant binding to phages displaying the beta mutant RBD and the delta mutant RBD; unfortunately, the VHH screened here failed to show the binding ability to the omicron mutant RBD.

Although the most effective NAbs often directly interfere with the binding of the RBD and ACE2 [32,42], which is also the original intention of this research to develop a competitive panning method, the neutralization ability of these NAbs drops sharply or even disappears for the omicron mutants. Some amino acid mutations in the RBD directly affect the binding strength between the virus and the target cell receptor, which can affect the infectivity of the virus [43]. This is often the main reason why some mutant strains have stronger transmissibility and escape from vaccines and antibodies. Notably, the contact ratio of binding sites of the MAb DXP-604, ACE2 and the RBD is 85%, as revealed by cryo-EM structures, which means that if the mutation site of the mutant prevents DXP-604 from binding to the RBD, it will be difficult for the mutant to bind to ACE2 itself, inhibiting invasion and infection of human cells. Therefore, DXP-604 still maintains the neutralizing ability, with an IC50 value of 0.287 μg/mL, for the omicron mutant [32]. In contrast, the small size of Nbs appears to be a double-edged sword—they may be escaped by individual mutations (especially those mutations on RBM motif) or bind to hidden epitopes and thus act in ways other than directly blocking ACE2-RBD binding. Most RBD mutations (9 out of 15) found in omicron strains are located in the ACE2 RBM [40]; the highly variable Q493/N/E/R/Y in omicron [44,45] directly caused theVHH_5-05_ to lose its neutralizing ability. Fortunately, relevant studies [39,46,47] have shown that NAbs against S protein or hidden RBD sites, conserved neutralizing sites, non-ACE2-competitive sites, etc., still can inhibit the infection caused by the omicron mutant. As mentioned above, Ying et al. reported that the antibodies n3113 and n3130 [39] cannot compete with ACE2 for RBD binding but can neutralize SARS-CoV-2 by targeting a ‘‘cryptic’’ epitope located in the spike trimeric interface. Coincidentally, some of the lateral surfaces and the hidden positions inside the spike trimer are perfectly conserved in the omicron S protein, and previous studies have also revealed the existence of conserved epitopes on the SARS-CoV-2 RBD [48,49,50]. Based on this, the bispecific sdAb they designed showed a broad-spectrum neutralization effect on omicron variants.

## 5. Conclusions

In summary, our study provided an RBD-ACE2 competitive screening method and successfully yielded a target antibody with complex binding requirements. The synergistic use of Nbs screened by this method that competes with ACE2 and other Nbs against SARS-CoV-2 (such as those that recognize hidden epitopes) may be ideal. We also expect that the application of multiple methods and platforms and the discovery of antibody drugs can contribute to the construction of the “Great Wall of Immunity” of the human body and contribute to the suppression of the coronavirus pneumonia pandemic in the near future.

## Figures and Tables

**Figure 1 vaccines-11-00371-f001:**
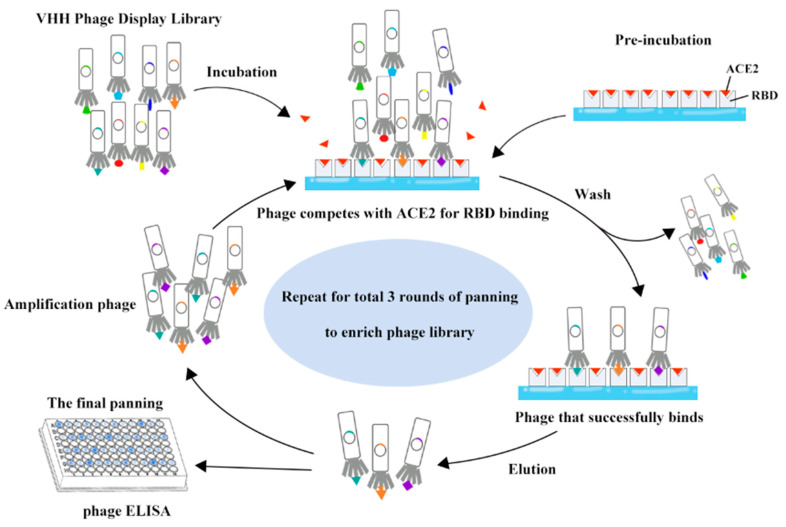
Method of the RBD-ACE2 competitive panning protocol.

**Figure 2 vaccines-11-00371-f002:**
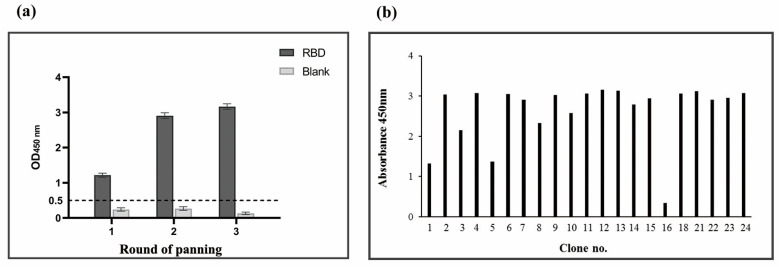
Identification of SARS-CoV-2 RBD-specific binders by polyclonal phage ELISA (**a**) or monoclonal phage ELISA (**b**).

**Figure 3 vaccines-11-00371-f003:**
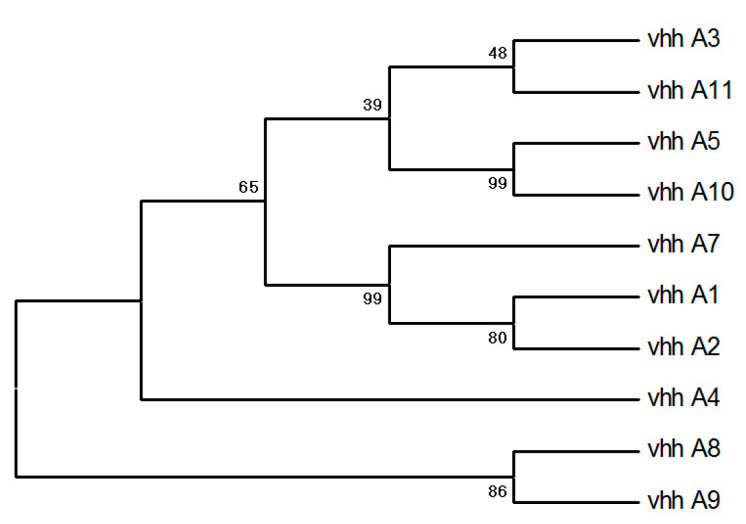
Phylogenetic tree of ten nanobodies.

**Figure 4 vaccines-11-00371-f004:**
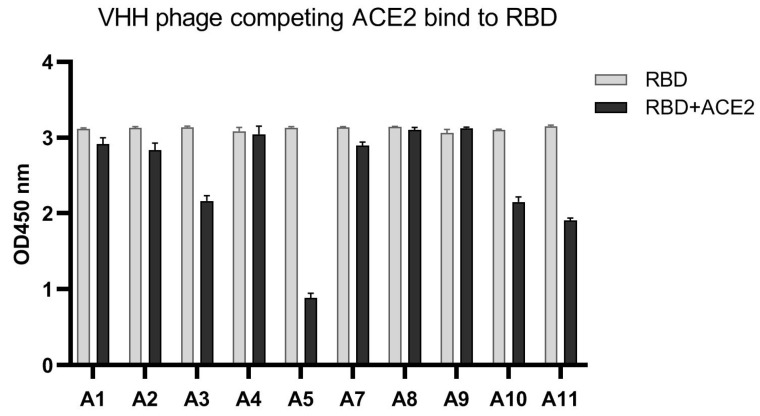
Competitive ELISA showed that 4 VHH-phages and ACE2 competed with each other to bind SARS-CoV-2 RBD.

**Figure 5 vaccines-11-00371-f005:**
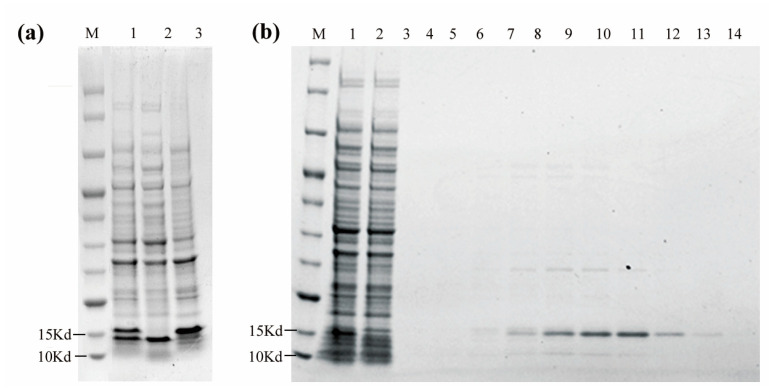
Analytical studies of prokaryotic expression and purification of VHH_5-05_. (**a**) Analysis of protein expression by reducing SDS-PAGE. Lane M, protein molecular mass marker; Lane 1, entire bacteria after induction; Lane 2 and 3, supernatant and precipitation after ultrasonic destruction of bacteria. (**b**) Analysis of protein purification by SDS-PAGE. Lane M, protein molecular mass marker; Lanes 1, supernatant after ultrasonic destruction of bacteria. Lanes 2, flow through; Lanes 3–14, gradient imidazole eluent.

**Figure 6 vaccines-11-00371-f006:**
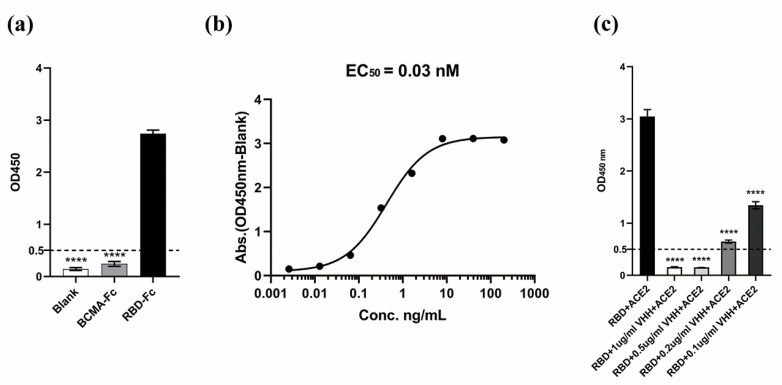
Validation of VHH activity. (**a**) Specific binding of VHH to the SARS-CoV-2 RBD. (**b**) Determination of the binding affinity of VHH. (**c**) The VHH competes with ACE2 for binding to RBD. The results are presented as the mean ± SD (*n* = 3), *p* < 0.0001 (****).

**Figure 7 vaccines-11-00371-f007:**
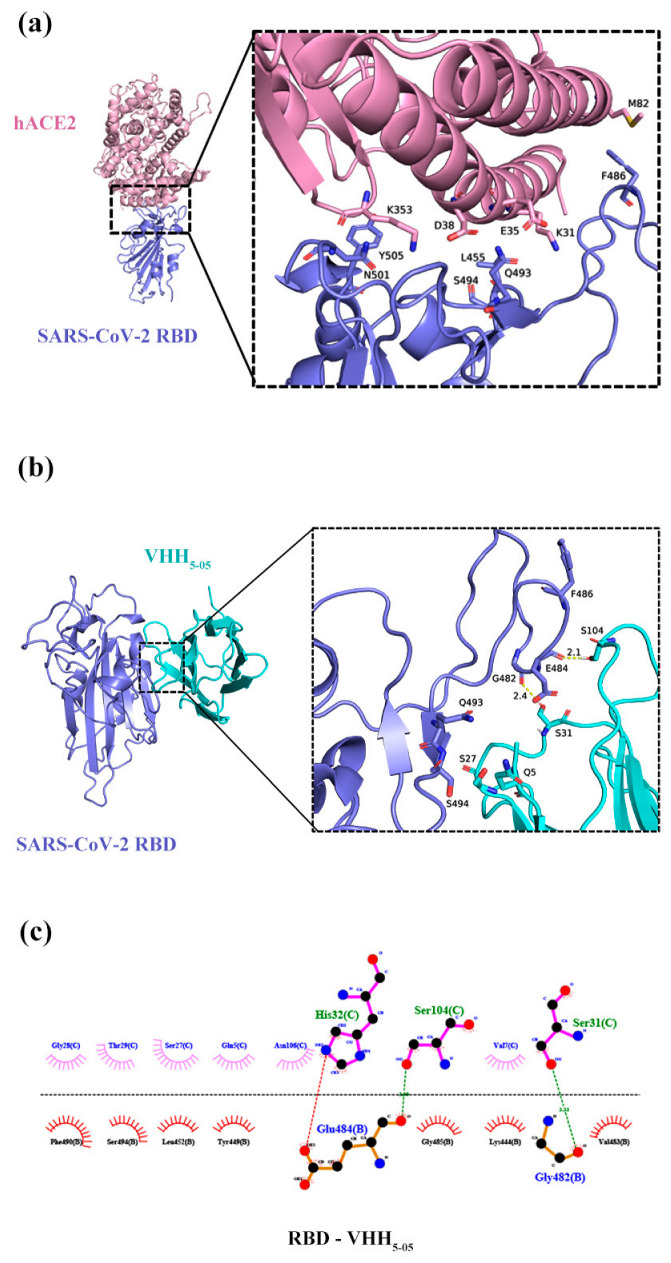
Structural analysis of the RBD-VHH_5-05_ and RBD-hACE2 complexes and comparison of their epitopes. (**a**) The crystal structure of the hACE2-SARS-CoV-2 RBD complex. The RBD and ACE2 are shown in blue and light pink, respectively. (**b**) Superimposition of VHH_5-05_ and RBD. VHH_5-05_ is represented in cyan. (**c**) 2D diagram of the complex.

**Figure 8 vaccines-11-00371-f008:**
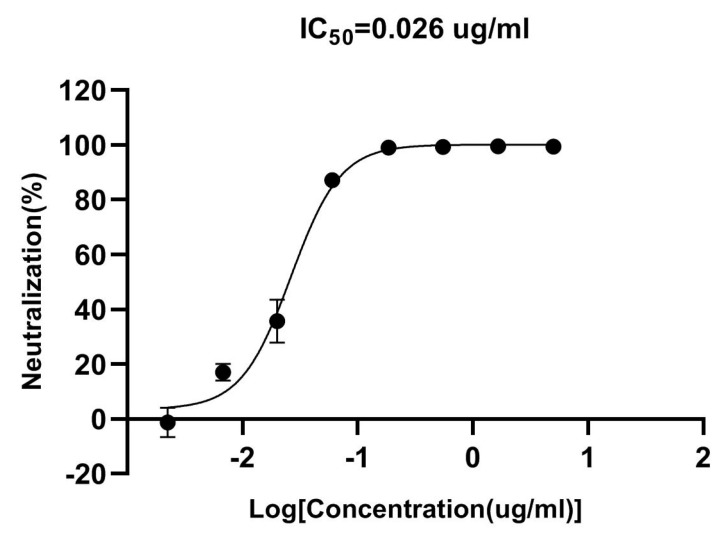
Pseudovirus neutralization of VHH_5-05_ against SARS-CoV-2. Pseudovirus assays were performed using 293/hACE2 cells. Data are shown as mean ± SD.

**Figure 9 vaccines-11-00371-f009:**
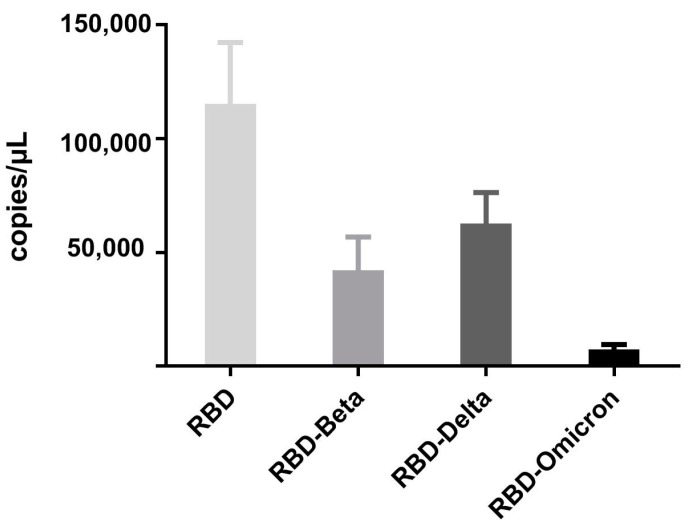
Analysis of the binding of the VHH to recombinant phages exhibiting the RBD of mutant strains of SARS-CoV-2. The results are the means of three independent determinations.

**Table 1 vaccines-11-00371-t001:** Selective enrichment of SARS-CoV-2 RBD-specific phages by three rounds of panning. PFU, plaque-forming units.

	RBD Concentration (µg/well)	ACE2 Concentration (µg/well)	Proportion of Tween-20 in PBST	Wash Times in Final Step	Input Titer (PFU)	Output Titer (PFU)	Recovery Efficiency	Fold Increase
Round 1	0.5	—	0.05%	10	1.4 × 10^11^	3.66 × 10^5^	2.6 × 10^−6^	—
Round 2	0.1	0.2	0.1%	20	3.31 × 10^11^	5.35 × 10^7^	1.6 × 10^−4^	62
Round 3	0.05	0.1	0.2%	30	2.93 × 10^11^	3.7 × 10^8^	1.3 × 10^−3^	8

## Data Availability

Not applicable.

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
