# Peer review of "A Competitive Panning Method Reveals an Anti-SARS-CoV-2 Nanobody Specific for an RBD-ACE2 Binding Site"

_vaccines, 2023, doi:10.3390/vaccines11020371_

Round 1

Reviewer 1 Report

The Manuscript ID: vaccines-2048873 is appropriate to the journal but it needs extensive English editing. Also the introduction difficult to understand. it should be improved. Results need to explained more and they should connect well with citations in discussion part. 

Reviewer 2 Report

I have read with great interest the article entitled "The discovery of an anti-SARS-CoV-2 nanobody specific for an 2 RBD-ACE2 binding site using a competitive panning method". I found it a very interesting approach, accurate and with a great elaboration. The figures seem accurate and eloquent to me as well as the structural analysis.

Great job with hopeful conclusions that open up a range of therapeutic possibilities.

Reviewer 3 Report

In this manuscript, He et al. reported the discovery of an anti-SARS-CoV-2 nanobody clone that is specific for RBD-ACE2 binding site using a competitive panning method. The authors claimed that by applying the ACE2 competitive panning, they could identify nanobodies that could specifically bind to SARS-CoV-2 RBD with high affinity and could successfully block the RBD-ACE2 binding. The authors also showed some structural simulation of the RBD-VHH complex to support the mechanism that they proposed. At the end of the study, the authors showed the binding of the newly identified nanobody clone in binding of the spike proteins from various SARS-CoV-2 variants including the latest Omicron variant.

Major issues:

1.     It is hard to understand the aims and objectives of the current study. In this manuscript, the author did not state clearly how this anti-SARS-CoV-2 RBD Llama antibody phage library was constructed. If we assume that a llama immunization process using SARS-CoV-2 WT RBD recombinant proteins has been done, it is not surprising to identify a VHH clone against the target with high affinity, even by applying the traditional bio-panning method. If the authors want to report a novel method which is superior than the traditional method, they should provide some comparison data to support their claim. Is it the truth that by using the traditional method, they failed to identify this clone? On the contrary, if the authors just want to claim that it is a faster and easier method to identify a good clone or like what they claimed in the manuscript (Pg 11, line 389-391) - “the original intention of this research is to develop the method as the neutralization ability of these Nabs drops sharply or even disappears for the omicron mutant”, they should not use the WT spike RBD but instead use Omicron spike RBD, which is more relevant to the current status of the pandemic.

2.     In section 3.2, the authors have claimed that they have identified 21 phage clones from the ELISA results and selected 10 monoclonal samples for sequencing analysis. How many unique sequences have been identified from the 10 samples? The authors just claimed that they randomly selected one sequence for further prokaryotic expression and characterization. What is the rationale in this selection? Would other clones also showed functions and high affinity? If the selection is not random and this is the only clone showing high affinity and potent blocking, whether this competitive panning method is better than traditional bio-panning and screening may be questionable. The authors should claim that it is a sound method only if they can demonstrate that most clones are tight binders and can block RBD-ACE2 binding.

3.     In figure 4, the authors showed specific binding of the selected VHH clone, the affinity binding ELISA result and the competitive binding to RBD using ACE2 blocking ELISA. However, they have not shown any actual functional characterization data such as pseudovirus neutralization assay. Blocking does not always mean neutralization. Some clones may block the ACE2-RBD binding without showing any neutralization activities. If this VHH clone only showed activity of blocking ACE2-RBD binding but cannot neutralize the SARS-CoV-2 virus, it has very limited application.

4.     In Figure 6, the authors used recombinant phages exhibiting RBD of different mutant strains of SARS-CoV-2, we have seen similar results as most other anti-SARS-CoV-2 antibodies which lost binding to the RBD of various VOCs, especially Omicron strain. Hence, even if the identified VHH has very high binding affinity, it still could not overcome the limitations brought by the multiple mutations of the virus. Again, back to the questioned raised in point 1, if the authors are able to use Omicron RBD in the panning process, the study could be meaningful.

Minor issues:

1.     In Table 1, the authors showed a column “Recovery Efficiency”, which is not necessary. Instead, the authors can list other items such as “ACE2 protein used in each round”, details of the washing step, etc. If the authors want to showcase their competitive method, they should refine Table 1 to provide clearer guidance to the readers.

2.     Please state clearly how the immunization steps were done as it is highly relevant to the current study. What proteins were used for the immunization?

Round 2

Reviewer 3 Report

I have browsed through the authors' responses to my comments. They have tried to address most of my queries by providing extra data (e.g. pseudovirus assay and phylogenetic tree of 10 nanobodies). Also, the authors provided some explanation on why this work still focused on WT strain and why only one VHH clone was chosen for further characterization. It may be understandable that the authors have limited resources and can only pick one clone. 

With all these necessary changes, it is now acceptable for publication. 

Author Response

Thank you very much for your encouraging response,now we hope the revised manuscript meet the journal’s standard and suitable for publication.